# Study on flow field characteristics of gas-liquid hydrocyclone separation under vibration conditions

Xiaoguang Zhang[1,2], Fan Yu[1,2], Yu Jin[1,2], Lixin Zhao[1,2], Suling Wang[1], Baorui Xu[1,2]*

1 School of Mechanical Science and Engineering, Northeast Petroleum University, Daqing, Heilongjiang, China, 2 Heilongjiang Key Laboratory of Petroleum and Petrochemical Multiphase Treatment and Pollution Prevention, Daqing, Heilongjiang, China

* xubaorui2009@126.com

**Data Availability Statement:** All relevant data are within the manuscript and its Supporting Information files.

**Funding:** The authors gratefully acknowledge the support of the China Postdoctoral Science

## Abstract

The complex vibration phenomenon occurs in the downhole environment of the gas-liquid hydrocyclone, which affects the flow field in the hydrocyclone. In order to study the influence of vibration on hydrocyclone separation, the characteristics of the flow field in the downhole gas-liquid hydrocyclone were analyzed and studied under the condition of vibration coupling. Based on Computational Fluid Dynamics (CFD), Computational Solid Mechanics Method (CSM) and fluid-solid coupling method, a fluid-solid coupling mechanical model of a gas-liquid cyclone is established. It is found that under the condition of vibration coupling, the velocity components in the three directions of the hydrocyclone flow field change obviously. The peak values of tangential velocity and axial velocity decrease, and the asymmetry of radial velocity increases. The distribution regularity of vorticity and turbulence intensity in the overflow pipe becomes worse. Among them, the vorticity intensity of the overflow pipe is obviously enhanced, and the higher turbulence intensity near the wall occupies more area distribution range. The gas-liquid separation efficiency of the hydrocyclone will decrease with the increase of the rotational speed of the screw pump, and the degree of reduction can reach more than 10%. However, this effect will decrease with the increase of the rotational speed of the screw pump, so the excitation effect caused by the rotational speed has a maximum limit on the flow field.

## 1 Introduction

At present, the water content of oil well produced fluid is high, which will lead to high cost of oilfield development and environmental pollution [1–3]. At the end of the 20th century, the Canadian Frontier Engineering Research Center first proposed the same-well injection and production technology (SWIP), which can realize the simultaneous completion of oil-water separation, low-water crude oil lift, and purified water injection in one oil well, which can effectively reduce ground water production and water treatment costs [4, 5]. Through the early application of SWIP, it is found that in the absence of gas, the technology shows good separation performance. However, with the later application, it is found that when the gas content in

Foundation (No.2021M690594), National Natural Science Foundation of China Joint Fund Project (U21A20104), Guiding innovation fund project of Northeast Petroleum University (No.2021YDL-14, 2022YDL-07), Research Start-up Fund of Northeast Petroleum University (No.2020KQ19).

**Competing interests:** The authors have declared that no competing interests exist.

the well is high, the existence of gas will not only reduce the efficiency of oil-water separation, but also produce a "cavitation phenomenon", which will cause damage and fracture of the wellbore and threaten the safe operation of oil production [6].

The separation mechanism of common gas-liquid separation technology is mainly divided into gravity sedimentation separation, filtration separation, inertial separation, hydrocyclone separation and so on [7–9]. The separation principle of each separation technology is different, which also results in adaptability to different working conditions. Among them, the hydrocyclone separation uses the density difference of different phases for centrifugal separation. Due to the advantages of low maintenance cost and simple structure, it is more suitable for being applied to small space environments such as downhole [10]. Therefore, many scholars have carried out a lot of research on gas-liquid hydrocyclone separation, and have obtained rich research results in terms of structural parameters, optimization of operating parameters, and flow field characteristics [11–14]. However, most of these studies are based on stable flow fields, and there are few studies on fluid-solid coupling in downhole hydrocyclone separation under excitation. However, the downhole environment is extremely complex. Under the interaction of various factors such as pump rotation, valve opening and closing, formation pressure and temperature changing with depth, each unit of the downhole system will produce complex vibration phenomena [15–17], and the resulting vibration will inevitably have an unpredictable impact on gas-liquid separation. Li S et al. [18] compared and analyzed the velocity field, pressure field and turbulent kinetic energy distribution in the variable diameter circular tube of a hydrocyclone under coupled and uncoupled conditions, and found that the influence of coupling on the flow field in the structure cannot be ignored. Xu Y et al. [19] studied the fluid-solid coupling of the spiral flow in the cyclone separator under vibration conditions, established a two-way fluid-solid coupling model of the cyclone under vibration conditions, and found that the movement of the structure caused the migration of the flow field structure. Li S et al. [20] found that the separation efficiency of the hydrocyclone under vibration coupling conditions is lower than that under uncoupled conditions. Therefore, it is particularly important to carry out the study of fluid-structure coupling under external excitation.

In this paper, the flow field characteristics of gas-liquid hydrocyclone under the action of single screw pump in SWIP are analyzed and studied. Based on the structure and working characteristics of the gas-liquid hydrocyclone on the vibrating wall, Computational Fluid Dynamics method (CFD), Computational Solid Mechanics method (CSM) and fluid-structure coupling theory, a fluid-structure coupling mechanical model is established. Using the fluid-structure coupling analysis method, the flow field characteristics such as velocity field, phase distribution, vorticity, and turbulent intensity in the downhole gas-liquid hydrocyclone under vibration conditions are analyzed. The influence law of vibration on gas-liquid separation provides a theoretical basis for the application of gas-liquid hydrocyclone separation in SWIP.

## 2 Models and methods

### 2.1 Structure design and working principle of gas-liquid hydrocyclone

The three-dimensional structural model and separation principle schematic diagram of gas-liquid hydrocyclone are shown in Fig 1. As shown in Fig 1A), the gas-liquid hydrocyclone is mainly composed of an inlet, an outer cylinder, a spiral flow channel, a swirl chamber, an inner shaft, gas phase outlet and liquid phase outlet. The working process of gas-liquid hydrocyclone is as follows: Firstly, the gas-liquid mixture enters the interior of the hydrocyclone through the inlet and then flows into the spiral flow channel. In the spiral flow channel, due to the effect of spiral guidance, the gas-liquid mixed fluid undergoes passive forced rotation. Under the effect of forced centrifugal force difference, the gas-liquid phase begins to separate.

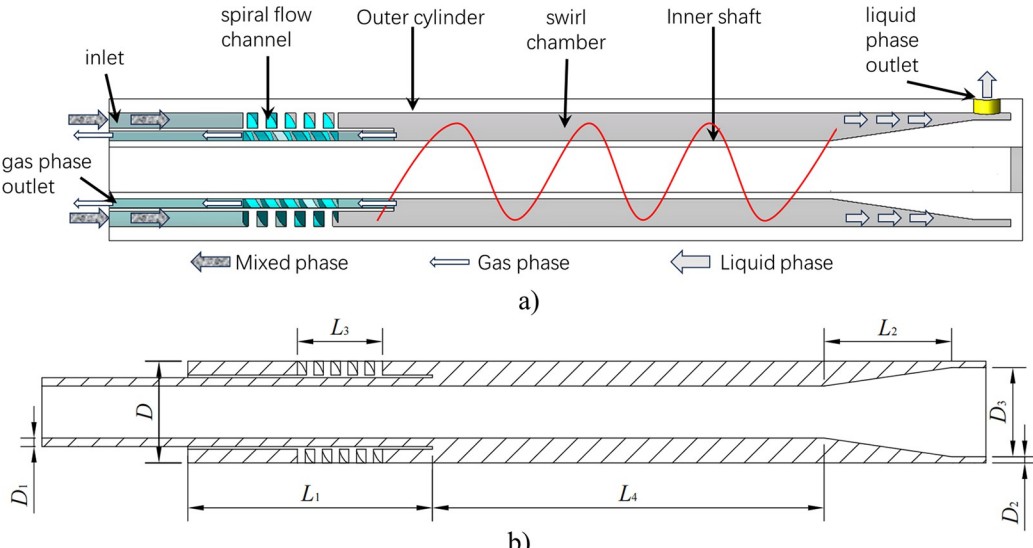

**Fig 1. Structural model of hydrocyclone.** a)Three-dimensional structural model and separation principle schematic diagram of gas-liquid hydrocyclone. b) Initial structure size of gas-liquid separator.

The gas phase with lower density bears less centrifugal force and flows towards the center area of the vortex, while the liquid phase with higher density bears more centrifugal force and flows towards the vicinity of the outer wall. The separation of gas-liquid phases occurs in the swirl chamber. In the swirl chamber, further stratified separation flow is formed between the gas and liquid phases, and after stable separation through a certain length of swirl chamber, the gas phase that converges to the center of the hydrocyclone flows out from the gas phase outlet, while the water phase that converges to the sidewall flows out from the liquid phase outlet. Finally, the separation of gas-liquid phases was completed.

The main size parameters of the gas-liquid separator structure are shown in Fig 1B). The main diameter $D$ is 96mm, the number of spiral guide blades is set to 5, the length of the gas phase flow channel is $L_1$, the inner diameter of the gas phase outlet is $D_1$, the height of the bottom flow inverted cone is $L_2$, the inner diameter of the liquid phase outlet is $D_2$, the pitch is $L_3$, the inner diameter of the bottom flow inverted cone is $D_3$, and the length of the main body of the separator is $L_4$. The main structural parameters of each part of the initial structure are shown in the Fig 1B), and the specific dimensions of the structural parameters of each part are shown in Table 1. Each parameter is expressed as a multiple of the main diameter $D$.

The wall of the swirl separation string is a rigid non-slip wall, and the wall of the swirl separation string is linearly elastic in a small deformation range. There is only mutual static friction

**Table 1. Main parameters of the initial structure.**

| parameter | value |
|-----------|-------|
| $D$/mm | 96 |
| $L_1$/mm | 2.4$D$ |
| $D_1$/mm | 0.083$D$ |
| $L_2$/mm | 1.25$D$ |
| $D_2$/mm | 0.0625$D$ |
| $L_3$/mm | 1.04$D$ |
| $D_3$/mm | 0.83$D$ |
| $L_4$/mm | 3.43$D$ |

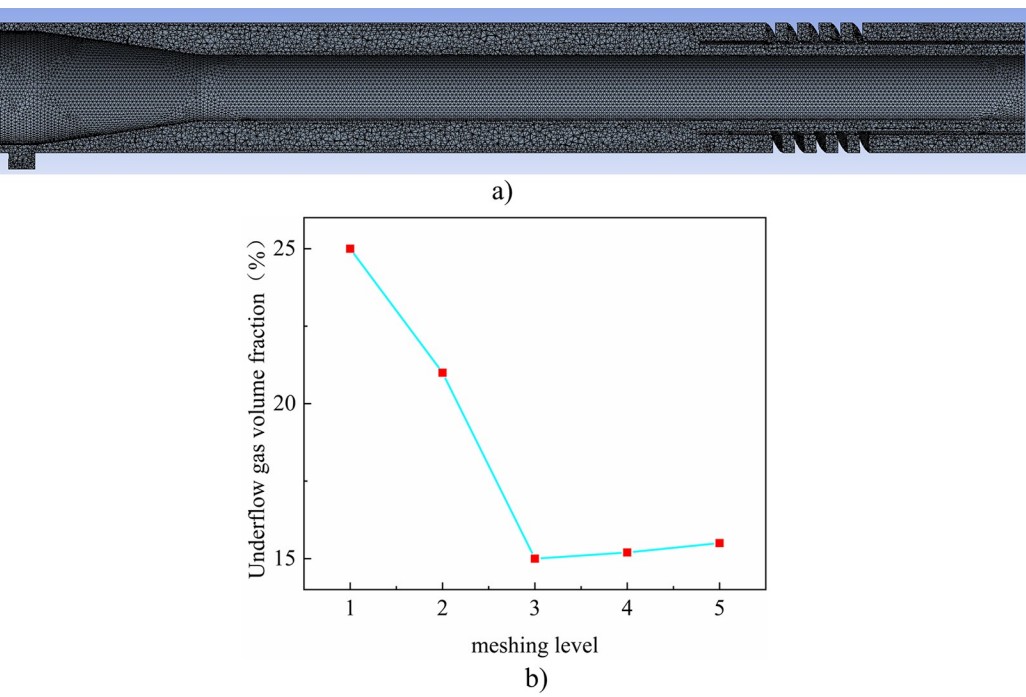

**Fig 2. Grid division of hydrocyclone.** a)Schematic grid division. b) Mesh independence verification.

between the fluid and the swirl separation string, and the rest of the damping is ignored. The mesh of the fluid domain is also divided by a tetrahedral mesh. The schematic diagram of the division results is shown in the Fig 2A), and the independence test of the mesh is carried out. The number of mesh elements corresponding to different division levels 1 to 5 is 650736, 846224, 1035930, 1255916, 1496220, respectively. The volume fraction of gas in the bottom stream can directly reflect the water removal performance of the gas-liquid separator, so the volume fraction of gas in the bottom stream is used as the test standard. The grid independence verification is shown in Fig 2B). The graphs corresponding to levels 3 to 5 show that the gas volume fraction of the underflow is relatively stable, and the final decision is based on the accuracy of the equilibrium numerical simulation and the calculation time, and the meshing level of 1035930 is adopted.

## 2.2 Fluid-structure coupling mechanical model

The ANSYS Workbench 2022R1 is used to establish the mechanical model of the downhole gas-liquid hydrocyclone, and the fluid is filled in the pipe string with the "Fill command". [21]. The interface where the structural domain and the fluid domain directly contact and interact is the fluid-solid interface [22]. In this paper, a twin-screw pump system is used to pump the outlet of the swirl separation string. In order to facilitate subsequent research, the following assumptions should be made:

1.  When the single screw pump is used, other units in the well work in an ideal state, and the exciting force on the swirl separation string is only generated by the recovery pump or the injection pump;

2.  The overflow and underflow connection pipelines are homogeneous and isotropic elastic material straight rods;

3. Ignore the influence of local stiffness and mass of flange joints on overall vibration;

4. The boundary conditions at both ends of the model are that one end is free and the other end is fixed;

5. The exciting force on the swirl separation string is a sinusoidal load that changes with time in one direction only in the radial direction.

The hydrocyclone selected in this paper works under the action of a single screw pump downhole, and only considers the excitation effect caused by the rotation of the screw pump. Therefore, the end connected to the screw pump at both ends of the hydrocyclone adds an excitation effect, while the other end is fixed. The established mechanical model is shown in Fig 3.

### 2.2.1 Fluid governing equation

The fluid microelements in the hydrocyclone should satisfy the conservation of mass and momentum. The governing equations of mass and momentum in the fluid domain are:

$$\frac{\partial \rho}{\partial t} + div(\rho u_i) = 0 \tag{1}$$

$$\frac{\partial \rho u_i}{\partial t} + \frac{\partial}{\partial x_j}(\rho u_i u_j) = -\frac{\partial p}{\partial x_i} + \frac{\partial}{\partial x_j}\left(\mu \frac{\partial u_i}{\partial x_j} - \rho \overline{u_i' u_j'}\right) \tag{2}$$

In order to more accurately simulate the characteristics of the gas-liquid separation flow field, a more accurate Reynolds stress model (RSM) is used to complete the turbulence calculation in the flow field. Pressure-strain sub-model adopts linear strain. The expression of the Reynolds stress transport equation is:

$$\frac{\partial}{\partial t}\left(\rho \overline{u_i' u_j'}\right) + \frac{\partial}{\partial x_k}\left(\rho \overline{u_k} \overline{u_i' u_j'}\right) = D_{ij} + P_{ij} + \varphi_{ij} + \varepsilon_{ij} \tag{3}$$

**2.2.2 Dynamic equation of tubular string structure.** The motion equation of the structure is that in the fluid-structure coupling analysis, the finite element method is used to

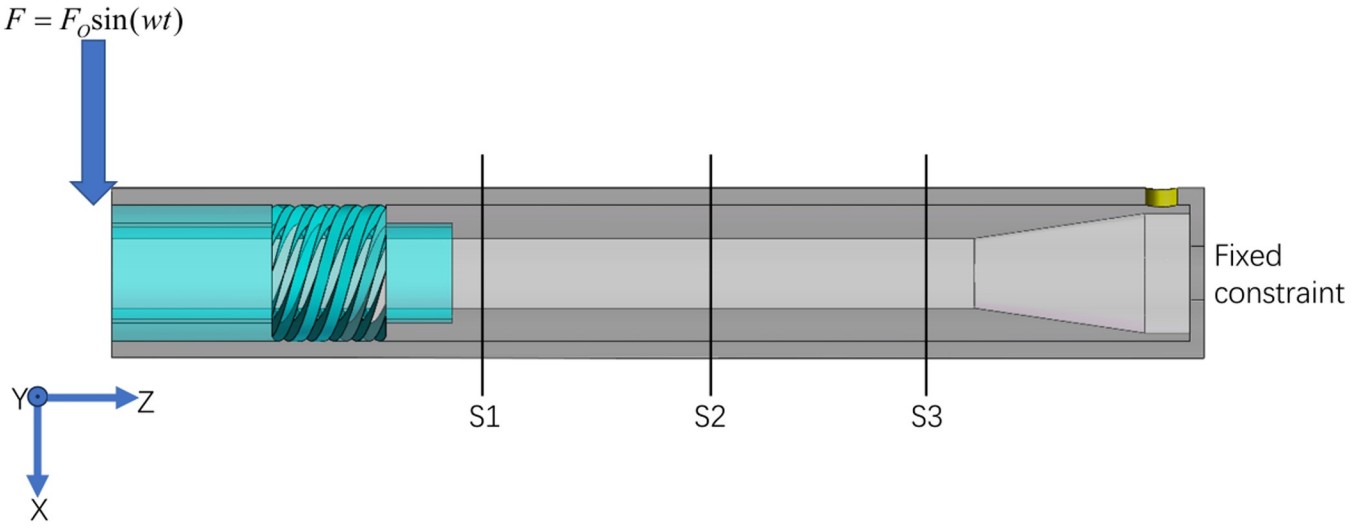

**Fig 3. Mechanical model.**

discrete the structure, considering the joint action of the internal and external forces of the pipe string structure, and the dynamic equation of the structure is:

$$[M]\{\ddot{\delta}\} + [C]\{\dot{\delta}\} + [K]\{\delta\} = \{F_f\} + \{F_o\} \tag{4}$$

## 2.3 Basic parameters and settings

The densities of water, oil, and gas that make up the mixed fluid are 998kg/m$^3$, 890kg/m$^3$, and 1.29kg/m$^3$, respectively, with volume fractions set at 0.65, 0.05, and 0.3. The turbulent flow model is the Reynolds stress model (RSM), and the Coupled algorithm is used to couple the pressure and velocity. The structural domain adopts steel (Structural Steel) with a density of 7850kg/m$^3$, an elastic modulus of 210GPa, and a Poisson's ratio of 0.3. The source of excitation mainly comes from the inertial force caused by the rotation of the screw pump rotor. The limit working speed of the screw pump used in the practical application of the same well injection system studied in this paper is 60r/min and 150r/min. Here, the working speed is set to 100r/min, so the excitation frequency rate is determined to be 1.7Hz. The excitation force amplitude is 240N calculated by the Formula (1) in the literature [23].

The multiphase flow model uses the Mixture model loaded with PBM, and the coalescence kernel and fragmentation kernel functions of the PBM model both use the Luo function [24]. See Eqs (5) and (6) for the coalescence kernel and fragmentation kernel functions. The gas flowing into the inlet is divided into 6 groups according to the size of the bubble particles. In order to make the bubble particle size distribution closer to the actual situation (when the gas content is 30%), the bubble size distribution is set by referring to the literature [25]. The bubble size distribution used in this paper is detailed in Table 2.

$$a(V_i, V_j) = \omega ag(V_i, V_j)Pag(V_i, V_j)[m^3/\sec] \tag{5}$$

$$\Omega br(V, V') = K \int_{\xi_{\min}}^{1} \frac{(1+\xi)^2}{\xi^n}\exp(-b\xi^m)d\xi \tag{6}$$

## 2.4 Boundary condition setting

The velocity inlet and the free outflow outlet are selected as the boundary conditions, the inlet velocity is 0.6m/s, the shunt ratio is set to 30%. The Coupled algorithm is used to couple the pressure and the velocity, the PRESTO method is used to discrete the pressure term [26], the wall function uses the standard wall functionand and the Time is selected as the transient. The difficulty of fluid-structure coupling lies in the calculation method of coupling solution and the processing of the dynamic grid. In this paper, the Fluent and Transient structural modules of Ansys are used to calculate the fluid domain and the solid domain, and the data interleaved iterative coupling is solved by the System coupling module. The spring smoothing and mesh

**Table 2. Bubble size distribution of the gas phase.**

| Number | Bubble size(μm) | Content |
|---|---|---|
| Bin-0 | 10 | 0.05 |
| Bin-1 | 6.35 | 0.15 |
| Bin-2 | 4 | 0.30 |
| Bin-3 | 2.52 | 0.30 |
| Bin-4 | 1.59 | 0.15 |
| Bin-5 | 1 | 0.05 |

**Table 3. Table for model input parameters and convergence criteria.**

| Type | Settings and valid values |
|---|---|
| **Fluid domain** | |
| Medium | water,oil,air |
| Density, kg/m$^3$ | 998(water); 890(oil); 1.29(air) |
| Volume fraction | 0.65(water); 0.05(oil); 0.3(air) |
| **Structural domain** | |
| Medium | Structural Steel |
| Density, kg/m$^3$ | 7850 |
| Elastic modulus, GPa | 210 |
| Poisson's ratio | 0.3 |
| **Operating parameter and convergence criteria** | |
| $n$, r/min | 100 |
| $T$, s | 0.59 |
| $f$, Hz | 1.7 |
| Excitation force $F$, N | 240 |
| Excitation displacement $X$, mm | $X = 240\sin(10.6*time)$ |
| Time step, s | 0.005 |
| Residual convergence standard | $1\times10^{-6}$ |

reconstruction methods are used to process the dynamic grid. The spring elastic coefficient is 0.6. QUICK scheme with second-order precision is used for the dispersion of momentum equation, second-order upwind scheme is used for turbulent kinetic energy and dissipation rate [27], and the accuracy of the residual is controlled to 1x10$^{-6}$. Here, in order to ensure the time resolution and stability of the calculation and prevent the negative mesh of the dynamic mesh during the reconstruction process, the time step of the three modules is set to 0.005s, the analysis time is 2s, and the mesh update frequency is 1. The input parameters and convergence criteria involved in the calculation of this article are shown in Table 3.

## 2.5 Reliability verification

**2.5.1 Fluid-structure coupling method verification.** The reliability verification is carried out by using the previous research on the fluid-structure coupling of the hydrocyclone. Due to the lack of relevant reports on gas-liquid fluid-structure coupling research, the research method in this article refers to experimental data on oil-water two-phase fluid-structure coupling in published literature. To verify the reliability of the model selection, we used the same method as in the literature [28] for simulation and compared the simulation results with experimental data. The model used in the verification simulation is an oil-water separation hydrocyclone. The simulation results are shown in Fig 4. It can be seen that except for the point with a vibration frequency of 4.6Hz and an amplitude of 1mm, the maximum error at all other points is less than 5%. Within the scope of the study, the separation efficiency of the simulation and experiment shows a consistent trend with the variation of amplitude, that is, both decrease with the increase of amplitude and frequency. This also indicates that the fluid-structure coupling simulation method in this paper is reliable.

**2.5.2 Validation of gas-liquid separation simulation method.** The accuracy of the gas-liquid separation simulation method is verified. Here, the separation efficiency experimental data from reference [29] was used as a reference for comparison with the results obtained from our simulation calculations using a multiphase flow model. The comparison results are shown in Fig 5. It can be seen that the average error between the simulated and experimental values of

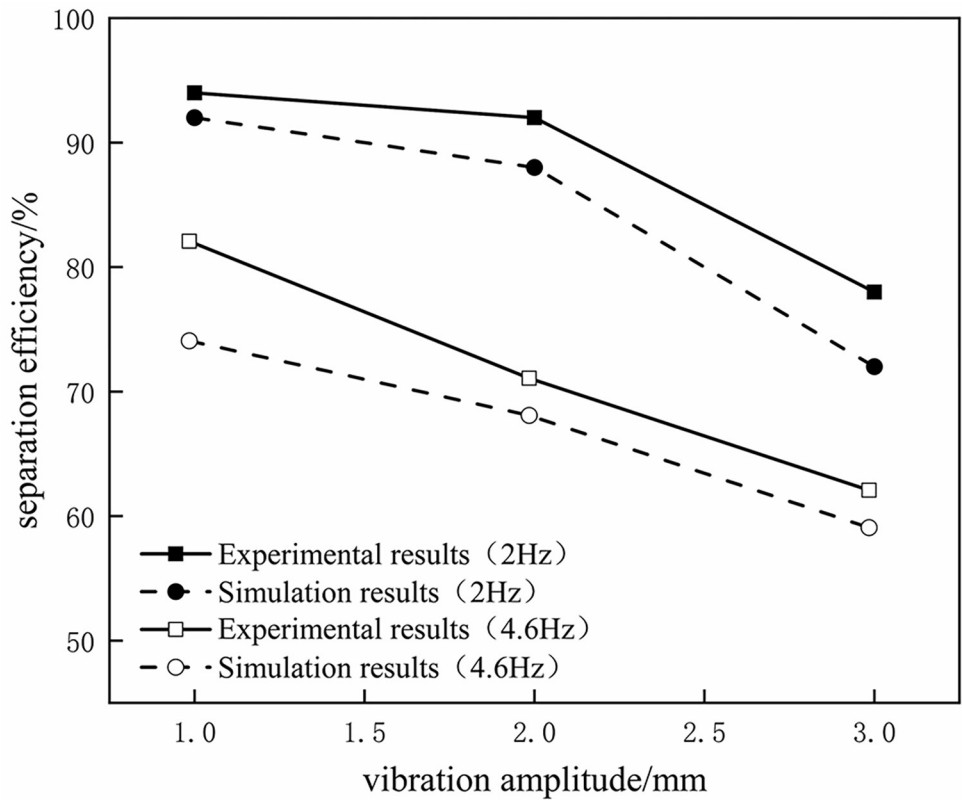

**Fig 4. Comparison curve between numerical simulation results and experimental results.**

gas-liquid separation efficiency is about 2.35%, and the separation efficiency increases with the increase of inlet velocity. The simulation results are basically consistent with the experimental results, indicating that the gas-liquid separation simulation method in this paper is reliable.

## 3 Results and analysis

### 3.1 Velocity field

Due to the high turbulence at the inlet, the flow field is relatively chaotic, and the swirl chamber is the main area for separation. Therefore, this article selects three cross-sections located far away from the inlet and in the vortex chamber to study their velocity field changes under external excitation conditions. The specific positions of the three sections are shown in Fig 3. Here, the displacement of the three sections in the excitation direction of the structure is extracted under the steady flow state after 1s. The results are shown in Fig 6. It can be seen that the S1 section curve value is the largest at the same time point, the S3 section curve value is the smallest, and the curve values of the three sections change at almost the same frequency, which is equal to the excitation force change frequency. Since the displacement curve value is the largest at 1.325s, 1.615s, 1.91s, and 2.505s, it is predicted that it will have a great impact on the flow field. In addition, from the analysis of the convergence curve in the simulation process, it can be seen that the simulation state of 2.505s is basically convergent. Therefore, the flow field of 2.505s is chosen for research below.

In order to study the effect of vibration coupling on the flow field of gas-liquid hydrocyclone, the distribution patterns of tangential velocity ($U_{tan}$), axial velocity ($U_{ax}$), and radial

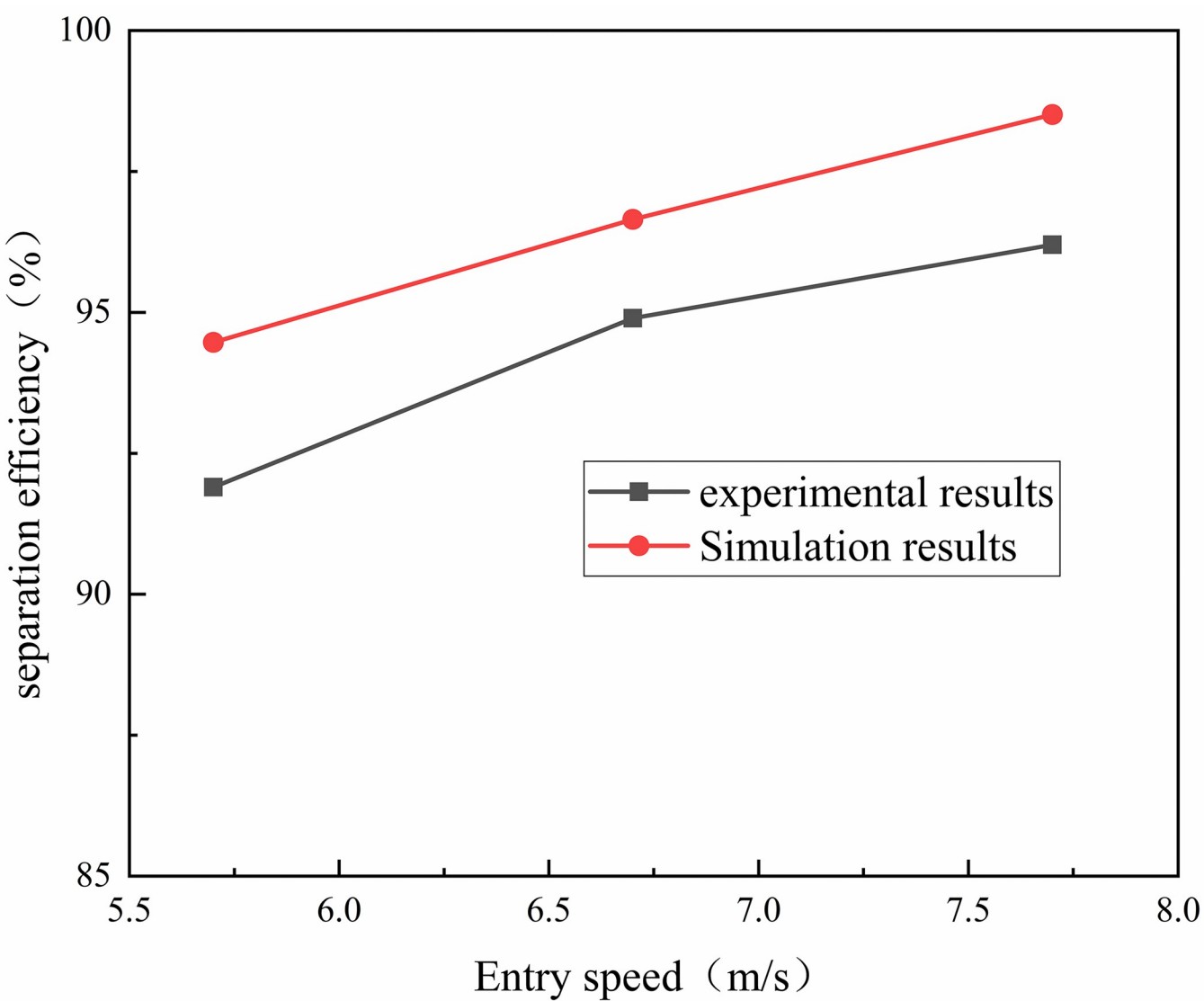

**Fig 5. Comparison curve between numerical simulation results and experimental results.**

velocity ($U_{rad}$) on three cross-sections under non-fluid-structure coupling (NO FSI) and fluid-structure coupling (FSI) were compared.

The results are shown in Figs 7–9 (in the figure, $U_{tan}/U_{in}$, $U_{ax}/U_{in}$ and $U_{rad}/U_{in}$ are the dimensionless tangential velocity component, dimensionless axial velocity component, and dimensionless radial velocity component, respectively. $U_{in}$ is the inlet velocity; $r$ is the radial distance, m; $R$ is the cross-sectional radius, m). Fig 7A) shows the distribution pattern of $U_{tan}/U_{in}$ on three different cross-sections under coupled and uncoupled conditions. It is seen that the overall trends of $U_{tan}/U_{in}$ of the three different cross-sections in the two cases are similar. The value of $U_{tan}/U_{in}$ under the coupling condition is significantly smaller than the value under the non-coupling condition. The tangential velocity is one of the main influencing factors affecting the separation. A decrease in tangential velocity will obviously affect the separation effect. It can be predicted that FSI will weaken the velocity intensity and centrifugal separation intensity in the swirl chamber. Fig 7B) shows $U_{tan}/U_{in}$ distribution in the X and Y

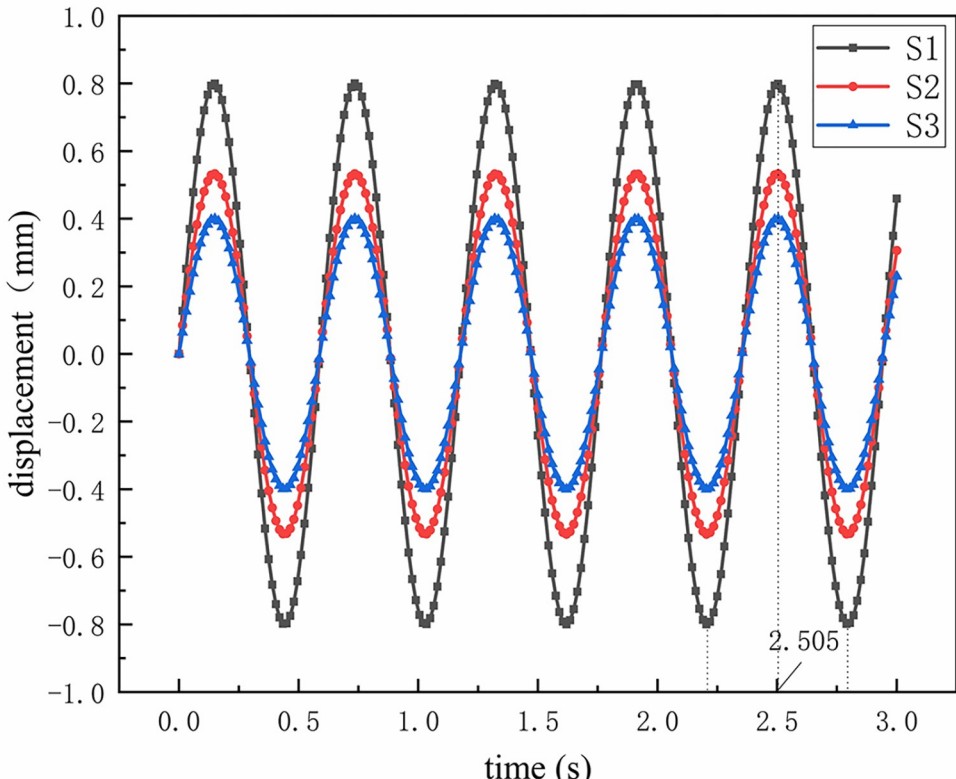

**Fig 6. The displacement of three cross-sections S1, S2, and S3 along the direction of structural excitation in the swirl chamber under stable flow state after 1 second.**

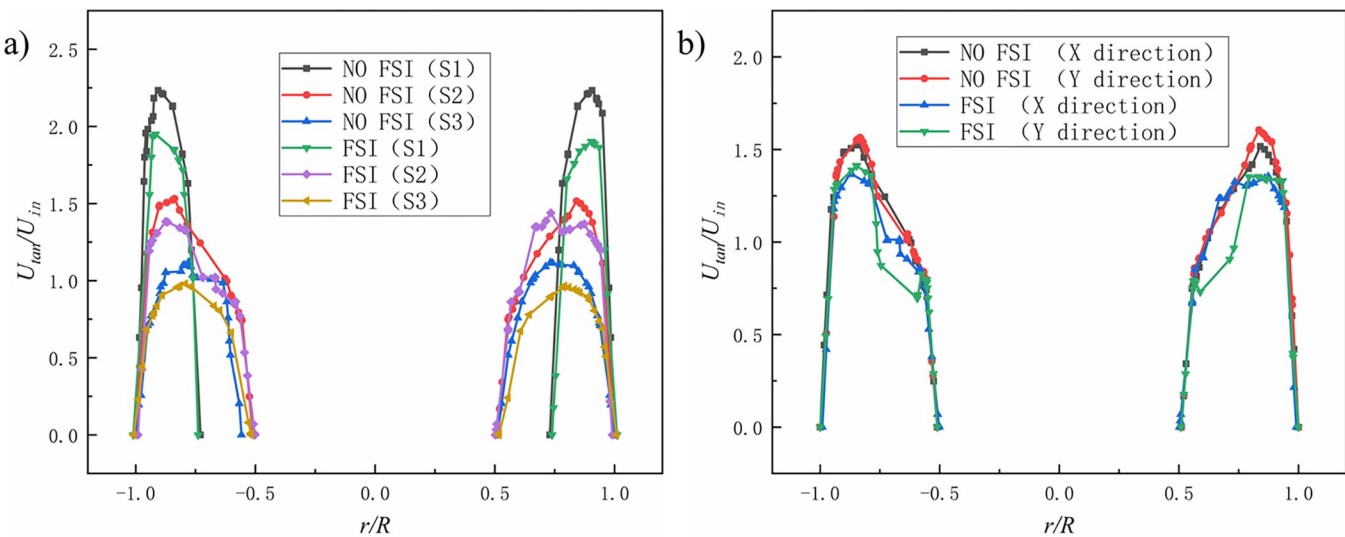

**Fig 7. Dimensionless tangential velocity($U_{tan}/U_{in}$) comparison.** a) $U_{tan}/U_{in}$ distribution of different sections; b) $U_{tan}/U_{in}$ distribution in X-axis direction and Y-axis direction of S2 section.

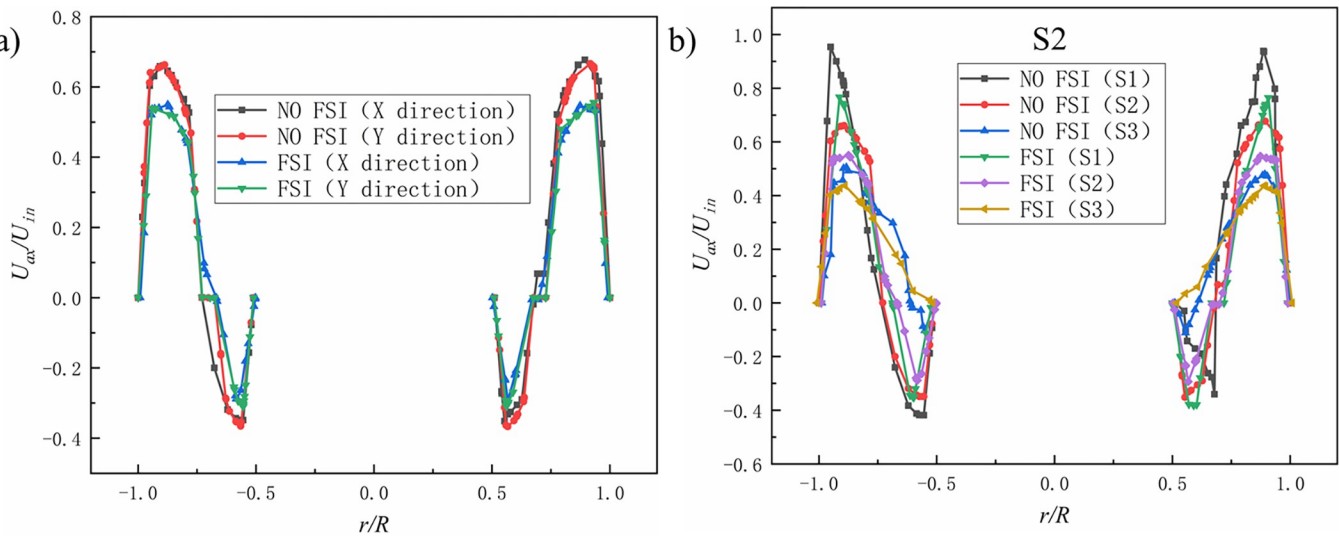

**Fig 8. Dimensionless axial velocity($U_{ax}/U_{in}$) comparison.** a)Axial velocity distribution of different sections; b) Axial velocity distribution in X-axis direction and Y-axis direction of S2 section.

directions of the S2 section in the case of FSI and NO FSI, where the X-axis direction is the excitation direction, and the Y-axis direction is the vertical direction. It can be seen that under the coupling effect, the peak of $U_{tan}/U_{in}$ in both the X-axis and Y-axis directions significantly decreases, and there is no significant displacement phenomenon in the Y-axis direction. $U_{tan}/U_{in}$ in the Y-axis direction is slightly higher than that in the X-axis direction, which may be due to the shear effect caused by the vibration displacement in the X-axis direction. There is not much difference in the overall distribution of $U_{tan}/U_{in}$ in the X-axis and Y-axis directions.

Fig 8A) shows the axial velocities at three cross-sectional positions under coupled (FSI) and uncoupled (NO FSI) conditions. It can be seen that the overall trend of axial velocity variation is similar in both FSI and NO FSI cases. The peak value of axial velocity near the central axis of the hydrocyclone is greatly reduced, which affects the upward exclusion of the central gas phase and thus has an adverse effect on gas-liquid separation. Fig 8B) shows the comparison of axial velocity distribution along the X-axis and Y-axis directions on the S2 cross-section under FSI and NO FSI conditions. The velocity distribution in the X-axis and Y-axis directions is basically the same, which is not conducive to the vertical exclusion of gas-liquid phases, and there is also a offset phenomenon in the X-axis direction.

Fig 9A) shows the radial velocities at three cross-sectional positions under coupled (FSI) and uncoupled (NO FSI) conditions. It can be seen that in FSI case, the radial velocity of the three cross-sections varies greatly. Among them, the S1 section has the most drastic changes, followed by the S2 section, and the S3 section has the smallest changes. The reason for this phenomenon may be the result of different excitation displacements of different cross-sections. Fig 9B) shows the comparison of radial velocity distribution along the X-axis and Y-axis directions on the S2 cross-section under FSI and NO FSI conditions. It can be seen that in FSI case, the distribution pattern of radial velocity along the X-axis and Y-axis is significantly different, and its asymmetry with respect to the central axis of the hydrocyclone is significantly intensified, which is obviously not conducive to the separation of gas and liquid phases.

By comparing the distribution patterns of three velocity components under FSI and NO FSI conditions, it can be inferred that the coupling effect will have adverse effects on the separation of gas-liquid phases. In order to investigate the migration of discrete phase bubbles in

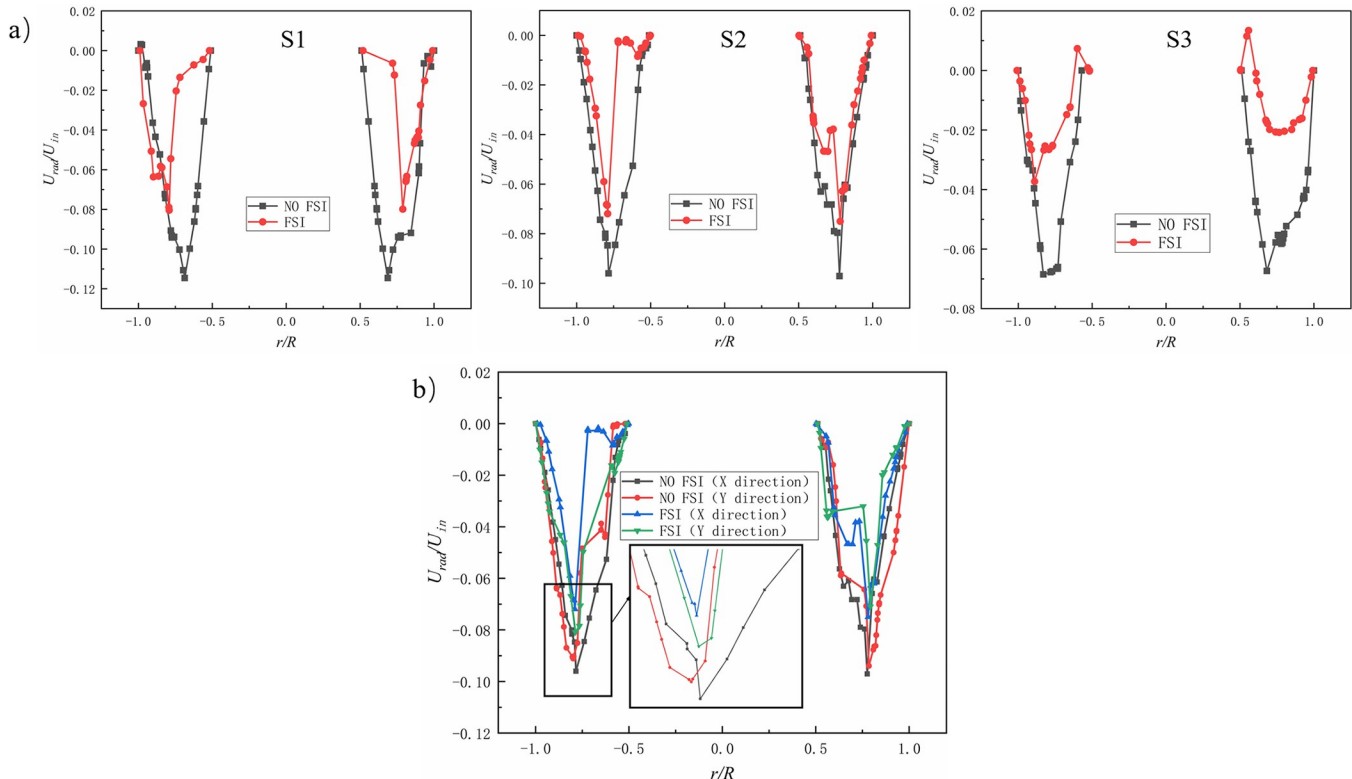

**Fig 9. Dimensionless radial velocity($U_{rad}/U_{in}$) comparison.** a) Radial velocity distribution of different sections; b)Radial velocity distribution in X-axis direction and Y-axis direction of S2 section.

the separation process more intuitively, two-dimensional velocity vector maps of the gas phase along the X-axis direction were extracted and compared under FSI and NO FSI conditions, as shown in Fig 10.

It can be seen that in NO FSI case, the gas phase exhibits a significant acceleration effect at the spiral flow channel, while in FSI case, the acceleration effect of the spiral flow channel on the gas phase is weakened, resulting in a smaller gas phase velocity value. In addition, in the case of NO FSI, within a long axial distance below the overflow pipe, the velocity of discrete phase bubbles points towards the gas phase outlet, while in the case of FSI, the velocity of most discrete phase bubbles points towards the liquid phase outlet. This phenomenon can also be explained by the influence of FSI on axial velocity. Observing the velocity distribution inside the gas phase outlet pipe section, it can be found that gas can be discharged smoothly from the gas phase outlet in the case of FSI, while in NO FSI, the gas phase is significantly hindered and even reflux occurs.

Therefore, based on the above analysis, it can be inferred that the excitation force will reduce the gas concentration at the gas phase outlet, thereby affecting the gas-liquid separation performance of the hydrocyclone. To further better understand this phenomenon, the influence of coupling on gas-liquid two-phase separation will be further explained from the distribution of other flow field related parameters.

### 3.2 Gas phase volume fraction

Fig 11 shows the distribution of gas phase. It can be seen that in FSI, the gas phase volume fraction at the gas phase outlet is relatively low, and the gas phase is more distributed in other

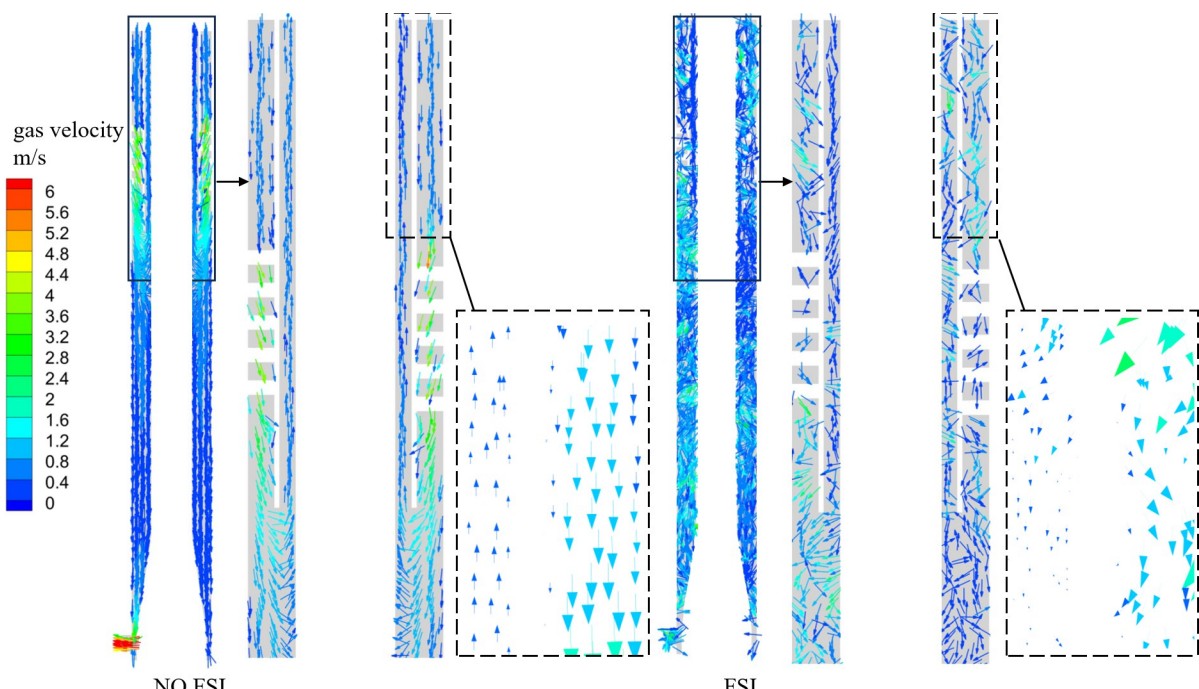

**Fig 10. Gas phase velocity vector diagram.**

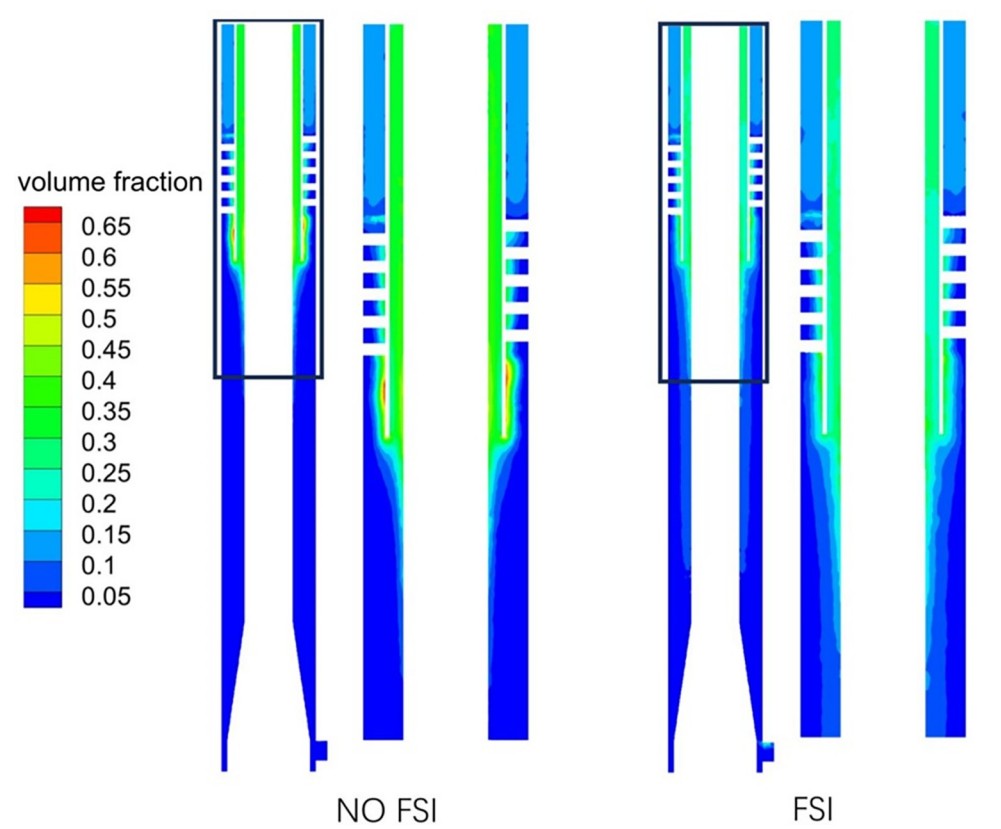

**Fig 11. Gas phase volume fraction distribution cloud map.**

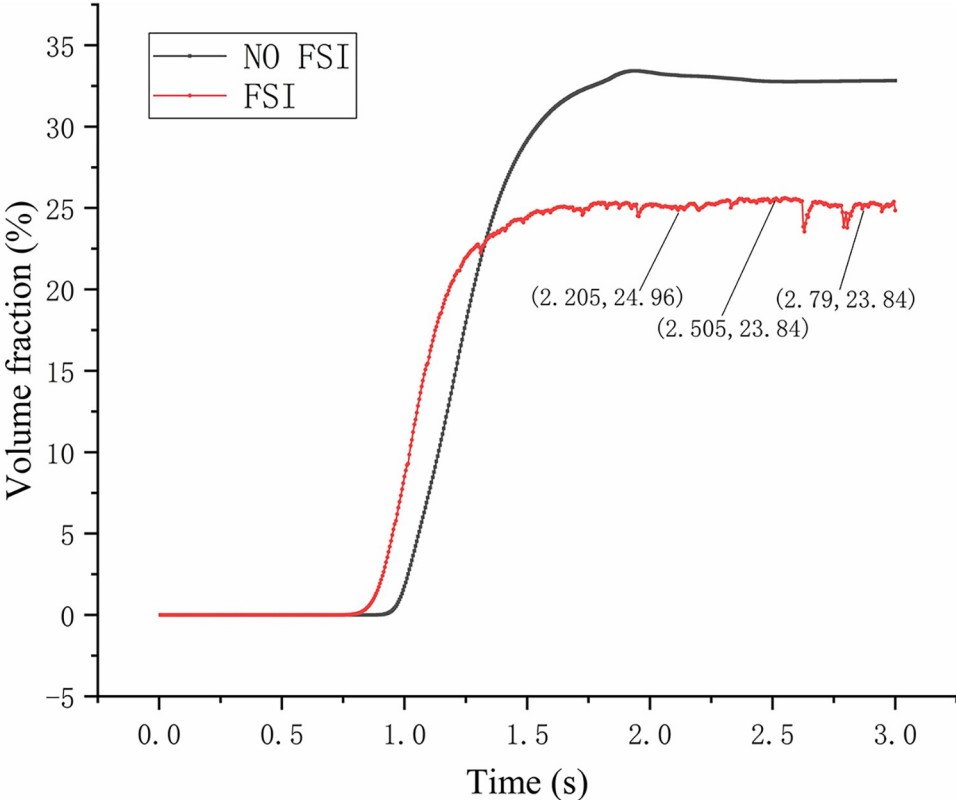

**Fig 12. Change curve of gas phase volume fraction at gas phase outlet.**

areas, indicating that the separation performance of gas-liquid phase is poor in this state. Fig 12 shows the monitoring curve of the change of the gas phase volume fraction at the gas phase outlet. By analyzing the convergence state of the simulated separation after 2 seconds, it can be seen that in NO FSI, the gas phase volume fraction curve at the gas phase outlet is relatively stable, while the gas phase volume fraction curve at the gas phase outlet considering FSI is clearly in a fluctuating state. Considering FSI, the fluctuation center value of the gas-phase volume fraction curve at the gas-phase outlet differs by about 10% from that NO FSI, indicating that the influence of excitation force on the flow field is a continuous and dynamic process.

In addition, by selecting the three time points corresponding to the maximum displacement curve in Fig 6: 2.205s, 2.505s, and 2.79, and matching these three time points with Fig 12, it can be found that the time point with the maximum displacement is not the time position with the lowest gas phase volume fraction. That is to say, there is a certain time difference between the effect of excitation force on displacement and the effect of excitation force on gas-phase separation flow field. Moreover, there is no clear pattern in the local curve fluctuation frequency of the gas phase distribution curve. It can be seen that considering the coupling effect of different time development, the flow field cannot be simply judged by the magnitude of displacement to determine its influence by the coupling effect. The influence process is more complicated and requires deeper research.

### 3.3 Vortex

Vortex is one of the important physical quantities representing the motion of vortices in a flow field. Fig 13 shows a cloud map of the distribution of vorticity intensity. It can be seen that the

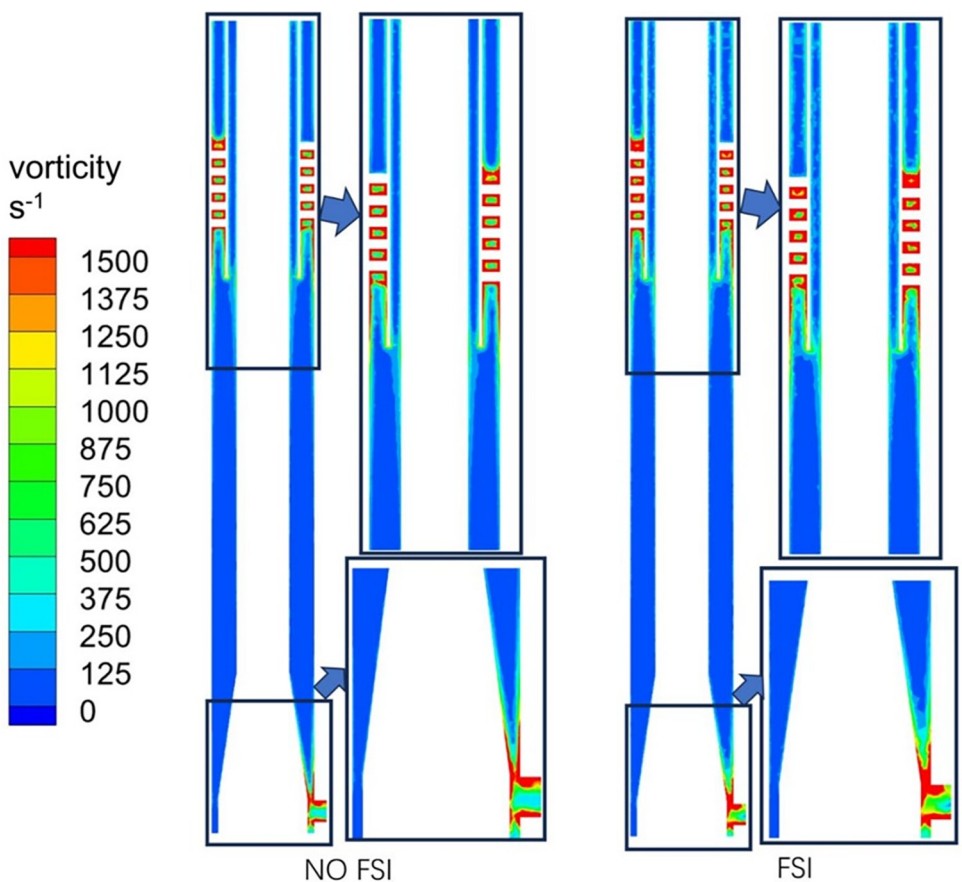

**Fig 13. Vorticity intensity distribution cloud.**

vorticity intensity distribution is different between FSI and NO FSI cases. In the case of FSI, the distribution pattern of vorticity field inside the gas phase outlet is poor, while for a hydrocyclone, the gas phase outlet is the gas discharge outlet, and the changes in the internal vorticity flow field will inevitably further affect the separation performance of gas-liquid two-phase. The above phenomenon is similar to the gas phase velocity vector distribution in Fig 10. The entire flow field under vibration conditions is relatively chaotic, especially in the gas phase outlet area, where gas emissions are affected and even backflow occurs.

## 3.4 Turbulence intensity

Turbulence intensity is an important parameter to describe the turbulence state, which can directly reflect the stability index of the flow field, and its value is the ratio of the turbulent fluctuation velocity to the average velocity. Fig 14 shows the cloud diagram of turbulence intensity distribution. It can be found that the turbulence intensity near the wall and the spiral flow channel is relatively large in these two states. The distribution phenomenon is consistent with the conclusion of conventional conditions. It can be seen that considering FSI will not change the basic distribution of turbulence intensity in the flow field. However, there are differences in the turbulence intensity distribution near the wall and gas phase outlet in both FSI and NO FSI cases. In the case of FSI, the turbulence intensity is relatively high, and the higher values in the flow field occupy more distribution areas.

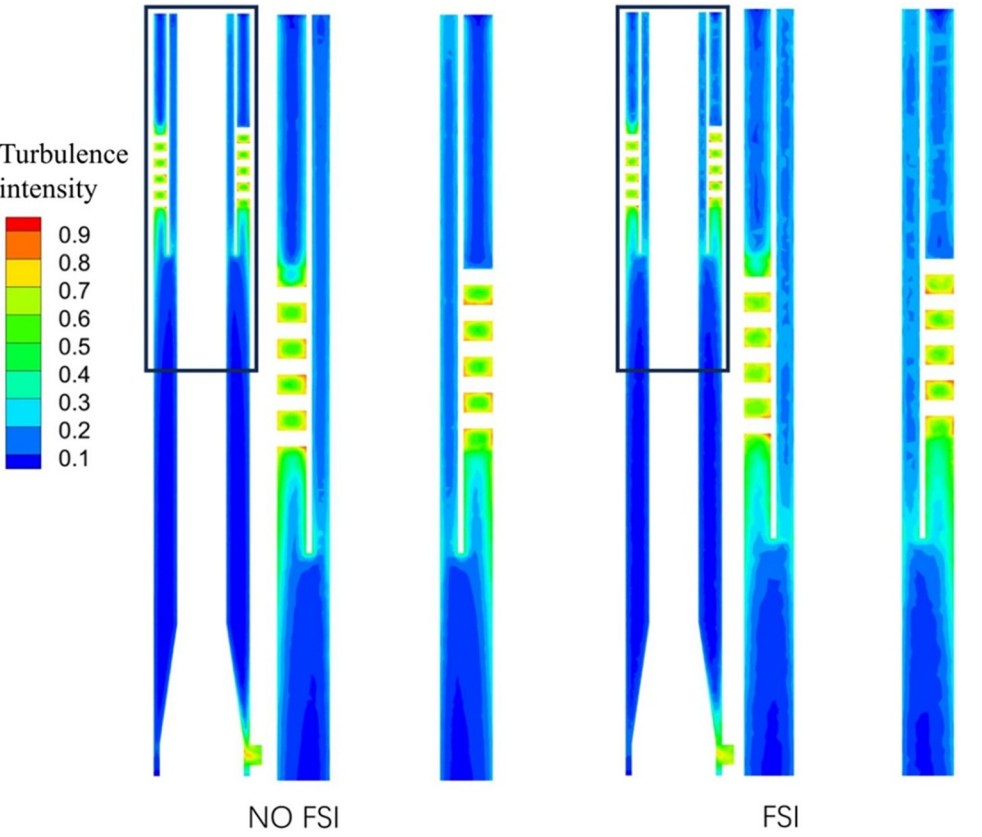

**Fig 14. Turbulence intensity distribution cloud.**

## 3.5 Bubble particle size

Fig 15 shows the distribution cloud of bubble particle size. In both FSI and NO FSI cases, the bubble particle size near the wall and spiral flow channel is relatively small. The reason for the small particle size of bubbles in the spiral flow channel may be due to the greater shear force generated by the spiral flow, which leads to bubble rupture. Combining Figs 14 and 15, it can be observed that the strong turbulence intensity in the gas-phase outlet pipe section in FSI case will have a crushing effect on the production of bubbles, and the distribution of bubbles in this area is relatively dispersed. In addition, by observing the area near the swirl chamber of the gas-phase outlet pipe section, it can be observed that bubbles with larger particle sizes have a larger diffusion area, indicating that large particle bubbles do not completely gather in the center under coupling conditions, but diffuse to other areas. That is, bubbles with larger particle sizes do not separate well, resulting in a lower gas volume fraction near the inner axis in the coupling state shown in Fig 11. The separation effect in this case is not ideal.

The tangential velocity is a key factor that provides power for gas-liquid separation in the gas-liquid separation process. Therefore, the distribution pattern of bubble particle size mentioned above can also be explained by the analysis of the tangential velocity component distribution in Fig 7: the tangential velocity in FSI case decreases significantly, causing bubbles to be unable to move higher towards the center, thereby reducing the separation effect. Fig 16 shows the cloud map of bubble particle size distribution in the cross-section of the gas-phase outlet pipe section at different times. It can be observed that the bubble particle size distribution changes over time, due to the unstable flow field caused by different displacements of vibration at different times.

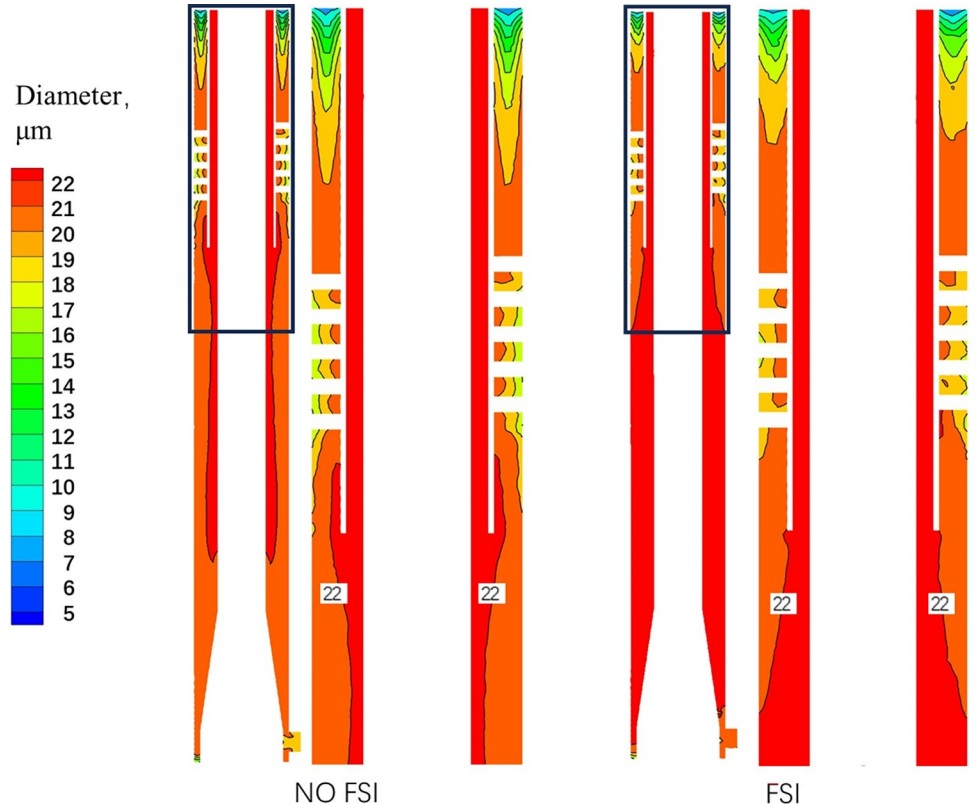

**Fig 15. The distribution cloud of bubble particle size.**

## 3.6 Separation efficiency at three speeds

From the above study of the characteristics of each flow field, it can be inferred that the influence of vibration on the flow field is comprehensive and variable. In order to further explore the influence of vibration on the separation performance of the cyclone, the separation efficiency is used to represent the separation performance of the cyclone under different vibration conditions. The separation efficiency is calculated by the overflow gas phase mass flow and the

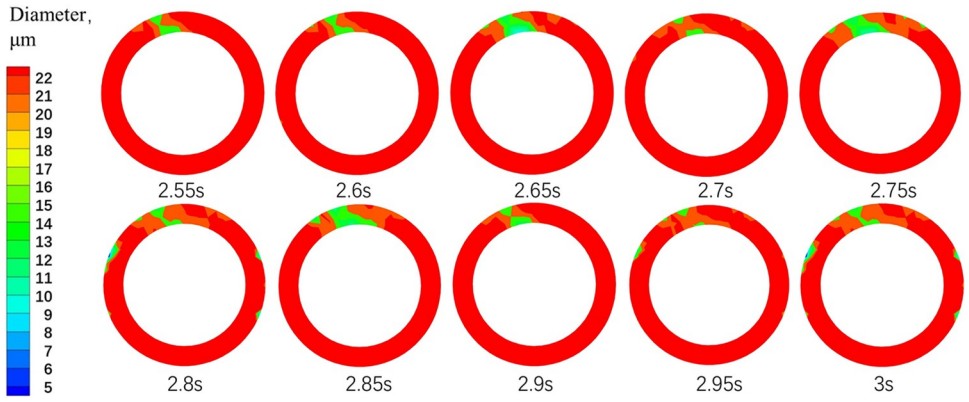

**Fig 16. Cloud map of bubble size distribution at different times of gas-phase outlet.**

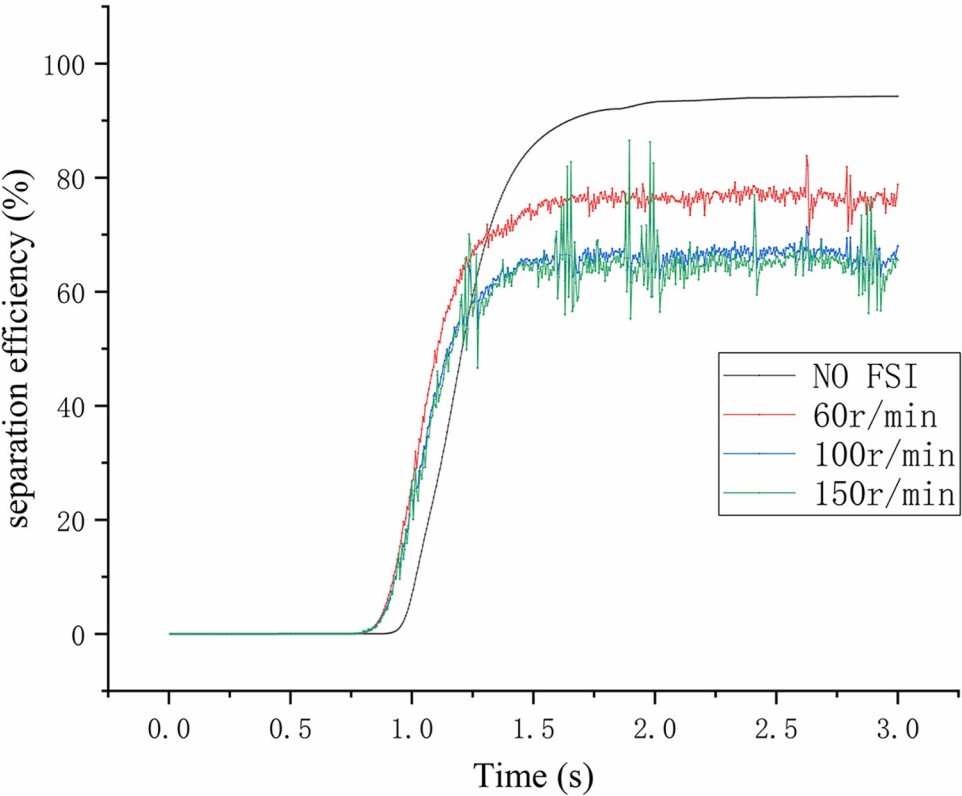

**Fig 17. Variation curve of separation efficiency.**

inlet gas phase mass flow, and the separation efficiency calculation formula is [27]:

$$E_z = \frac{M_{og}}{M_{ig}} \tag{7}$$

The maximum operating speeds of the screw pump mentioned in this article are 60r/min and 150r/min. Here, a comparison is added with the vibration effects of these two speeds. The excitation frequencies at these two operating speeds are determined to be 1Hz and 2.5Hz, and the calculated excitation force amplitudes are 80N and 520N, respectively, ignoring the flow rate changes caused by speed changes. The comparison of the monitoring curves of separation efficiency over time under three vibration conditions is shown in Fig 17. It can be seen that in FSI case, it takes less time for the three speeds to achieve stable separation. From this perspective, vibration may reduce the time for the entire system to reach a stable state. It should be noted that after the system reaches stability, the separation efficiency of NO FSI case is the most stable, which is a truly stable state. At this time, the separation efficiency basically does not change with time. However, unlike this, although the separation efficiency under the three rotational speeds also tends to a stable state, there are certain fluctuations in the curve, indicating that the flow field has been affected by the excitation force and is in a sub-stable state during this process.

Under the vibration conditions caused by three different screw pump speeds, the separation efficiency of the hydrocyclone decreases by more than 5% compared to FSI case. These changes are consistent with the gas phase volume fraction change curve at the gas phase outlet in Fig 12, and there is no direct relationship between the frequency and displacement of

vibration. In addition, the speed of the screw pump directly affects the degree of reduction in separation efficiency, with the minimum reduction in separation efficiency at 60r/min and the maximum reduction at 150r/min. However, during the process of rotating from 100r/min to 150r/min, the impact on separation efficiency weakens, and it can be inferred that the vibration caused by the screw pump speed has the maximum impact on the flow field.

## 4 Conclusion

In this paper, the fluid-solid coupling method is used to study the influence of screw pump vibration on the flow field of gas-liquid hydrocyclone from multiple perspectives such as velocity field, gas phase volume fraction, vorticity intensity, turbulent flow intensity, and gas particle size. Some internal connections are obtained, and the following conclusions are drawn:

1. The vibration coupling condition has a great influence on the three velocity components, and this influence is unfavorable to the gas-liquid separation: the tangential velocity and axial velocity of the three sections become significantly smaller near the peak value, and the X direction and the Y direction, the velocity change effect is basically the same; the different displacements of different sections will affect the degree of change of the radial velocity, and the radial velocity changes in the X and Y directions are quite different, but both aggravate the asymmetry of the velocity field.

2. Under the vibration coupling condition, the gas phase volume fraction near the overflow outlet is relatively low, and the separation effect of large particle bubbles is relatively poor without vibration conditions. The gas phase volume fraction and bubble distribution of the overflow outlet section change with time.

3. Under the vibration coupling condition, the distribution regularity of the vorticity inside the overflow pipe is poor, and the related flow field of the overflow pipe has a more obvious vortex enhancement effect; the turbulent intensity near the wall is relatively higher, and the area occupied by the higher value in the flow field The distribution range is also more, and the turbulence intensity distribution in the overflow pipe also becomes irregular.

4. The separation efficiency under vibration caused by the three rotational speeds is reduced by more than 10%. The larger the rotational speed, the lower the separation efficiency. However, this effect will weaken with the increase of the screw pump speed. The excitation effect caused by the rotational speed has a maximum limit on the flow field.

## Supporting information

**S1 Data.**
(XLSX)

## Author Contributions

**Conceptualization:** Baorui Xu.

**Data curation:** Xiaoguang Zhang, Yu Jin, Baorui Xu.

**Formal analysis:** Lixin Zhao.

**Investigation:** Xiaoguang Zhang.

**Methodology:** Xiaoguang Zhang, Baorui Xu.

**Software:** Xiaoguang Zhang, Yu Jin, Baorui Xu.

**Supervision:** Baorui Xu.

**Validation:** Xiaoguang Zhang, Baorui Xu.

**Visualization:** Fan Yu.

**Writing – original draft:** Xiaoguang Zhang, Fan Yu, Suling Wang, Baorui Xu.

**Writing – review & editing:** Xiaoguang Zhang, Baorui Xu.

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
