## [Decision Letter · Decision Letter 0]

16 Apr 2024

PONE-D-24-11282Study on flow field characteristics of gas-liquid hydrocyclone separation under vibration conditionsPLOS ONE

Dear Dr. Xu,

Thank you for submitting your manuscript to PLOS ONE. After careful consideration, we feel that it has merit but does not fully meet PLOS ONE’s publication criteria as it currently stands. Therefore, we invite you to submit a revised version of the manuscript that addresses the points raised during the review process.

As pointed out by the reviewers, the authors are suggested to provide details / dimensions of geometry and computational domain, include governing and relevant equations and mention discretization schemes.

The results of grid independence tests and comparison with other computational and experimental studies are also required.

The quality of Figures also need to be improved, in particular the velocity vectors (Fig. 9) so that flow pattern of gas and liquid becomes clear.

We look forward to receiving your revised manuscript.

Kind regards,

Muhammad Shakaib, PhD

Academic Editor

PLOS ONE

Journal Requirements:

“The authors gratefully acknowledge the support of the China Postdoctoral Science Foundation (No.2021M690594), National Natural Science Foundation of China Joint Fund Project (U21A20104) ,Guiding innovation fund project of Northeast Petroleum University (No.2021YDL-14), Research Start-up Fund of Northeast Petroleum University (No.2020KQ19).”

Please confirm at this time whether or not your submission contains all raw data required to replicate the results of your study. Authors must share the “minimal data set” for their submission. PLOS defines the minimal data set to consist of the data required to replicate all study findings reported in the article, as well as related metadata and methods (https://journals.plos.org/plosone/s/data-availability#loc-minimal-data-set-definition)

Authors do not need to submit their entire data set if only a portion of the data was used in the reported study

5. Please amend the manuscript submission data (via Edit Submission) to include authors Fan Yu, Yu Jin, Lixin Zhao, Wang Suling.

6. Please include a separate caption for each figure in your manuscript.

7. Please include your tables as part of your main manuscript and remove the individual files. Please note that supplementary tables (should remain/ be uploaded) as separate "supporting information" files

Reviewers' comments:

Reviewer's Responses to Questions

**Comments to the Author**

1. Is the manuscript technically sound, and do the data support the conclusions?

Reviewer #1: Partly

Reviewer #2: Yes

2. Has the statistical analysis been performed appropriately and rigorously? 

Reviewer #1: Yes

Reviewer #2: Yes

3. Have the authors made all data underlying the findings in their manuscript fully available?

Reviewer #1: Yes

Reviewer #2: Yes

4. Is the manuscript presented in an intelligible fashion and written in standard English?

Reviewer #1: Yes

Reviewer #2: Yes

5. Review Comments to the Author

Reviewer #1: This article studied the hydrocyclone under the influence of vibration by using computational fluid dynamics. However, it provides the several concerning points which are needed to address. So, in order to improve the article quality, the manuscript must be strictly revised by considering the following comments.

1) For abstract, the authors did not inform the reader that computational fluid dynamics was used as a tool for the authors’ work. So, the reviewer suggests the authors to add the comprehensive and informative abstract in the revised manuscript.

2) From Figure 1, the reviewer understands that the hydrocyclone is horizontally located. Does the reviewer understand your considered hydrocyclone correctly? Please clearly describe your considered hydrocyclone system in the revised manuscript.

3) The authors only showed the schematic of the considered hydrocyclone. However, there is no information of the dimensions of the considered hydrocyclone. Please report all details of the considered hydrocyclone in the revised manuscript.

4) The authors used Reynolds stress turbulence model for this work. However, the pressure-strain sub-model was not reported. Please report the pressure-strain sub-model in the revised manuscript.

5) The authors must also report the selected wall functions of the present work.

6) The authors only showed the information on an algorithm (Coupled) used in the present work. However, the information of spatial and temporal discretization schemes was absent. Therefore, in the revised manuscript, the spatial and temporal discretization schemes used in the present work must be reported. Moreover, please make the discussion between the selected set of numerical schemes of this work and the suggested numerical scheme set of the previous work (10.1016/j.powtec.2023.118713).

7) Please compare and show the residence time and the selected time step size of 0.005 s in the revised manuscript to confirm that the selected time step size is appropriate for the present simulations (Generally, the time step size should be shorter than 1/100 of residence time).

8) The authors did not show the convergence criterion for the present simulations. Please show the information on convergence criterion in the revised manuscript.

9) Please represent the governing equations used in the present work in the revised manuscript.

10) The authors used mesh-based CFD model for the present work. However, the details of grid generation and figure of grid generation used in the present work were absent. Please add information of grid generation of the considered hydrocyclone in the revised manuscript.

11) In the present work, the authors did not show grid independence study, which is an important section of the CFD article, in the present manuscript. The authors must add grid independence study in the revised manuscript.

12) For validation, the authors validated the model by considering the collection efficiency. However, the unknown parameters for the RANS equations are mean pressure and velocities. Hence, it is better to validate the present CFD model by comparing the simulated velocity profiles with the measured profiles. Please describe why the author considered collection efficiency for model validation or show the model validation by using the velocity profile comparison.

13) For velocity profile comparisons, the horizontal axis labels were dimensionless radial distance (r/R). At first sight, these horizontal axis labels are seemingly proper for velocity profile comparisons. However, the velocity profiles for swirl flow inside cyclone separators are not symmetrical. Therefore, the authors must use dimensionless distance (e.g., x/R for section S2 of Figure 2) for velocity profile comparisons in the revised manuscript.

14) Color map for contour of Figure 15 was not presented. Please add it into the Figure 15 of the revised manuscript.

15) Typing errors and English grammar must be checked.

Reviewer #2: In this study, the authors used external volume force as vibrational energy to improve gas-liquid separation efficiency. However, some comments are regarding the numerical modeling of multiphase flow in gas-liquid hydrocyclone device.

Comments are given below:

1. In the Introduction section, please cite the reference related to the Computational Fluid Dynamics method (CFD). Is it related to the VOF method?

2. In the introduction section, please cite the references related to the computational solid mechanics method, fluid-structure coupling theory, and fluid-structure coupling mechanical model.

3. In the models and method section, please mention the dimensions of the schematic figure in the Figure 1 caption.

4. In the fluid-structure coupling model, please mention the version of the ANSYS workbench that was used to perform the numerical simulation.

5. In the one-way coupling (where fluid dynamics are accounted for), please explain the major assumptions of the numerical simulations.

6. Please cite the reference of the "Fill command" that will be useful to readers.

7. Please cite the reference of "Fluid-Structure coupling interface."

8. It would be nice if the authors could explain all essential governing equations related to the mixture CFD model, kernel mixture model with PBM, and coalescence kernel, or the authors could cite the references related to the proposed numerical methods. In this physical problem layout, the essential governing equations could be split into two parts (i.e., FSI model (where fluid mechanics and solid structure mechanics are accounted for) and No FSI model (where Turbulence model: Reynolds stress model could be discussed). Based on the essential governing, authors could discuss the boundary conditions and numerical solver settings in the material and method sections or the appendix section.

9. In the material and method sections, authors could discuss the grid generation in terms of mesh independence study, model selection and boundary conditions, and solvers (i.e., phase bound simple algorithm), respectively, or this section could be placed in the appendix section, where, all essential governing equations will be discussed.

10. In the validation section, the authors validate experimental results with obtained numerical results because, in the context of the numerical methodology section, authors used water, oil, and gas as working fluid systems, but in the experimental section, authors used experimental results where two-phase (oil-water) fluid-structure interaction coupling is accounted for. We should validate similar systems.

11. In the basic parameters and settings section, authors could make a table related to all model input parameters with numerical solver settings (with convergence criteria).

12. In the velocity field section, it would be nice if the authors could discuss axial, radial, and tangential velocity distributions with streamlined velocity vector plots at different locations of gas-liquid hydrocyclone separators. These plots will be valid for the NO FSI model and the FSI model, respectively, and discuss the significant difference between the two proposed methods (i.e., NO FSI and FSI model).

13. In the bubble particle size section, authors could make a plot related to the separation efficiency as a function of gas bubble size with different inlet velocities of the gas phase and discuss the physical insight of the process.

14. In the vortex section, authors could make one plot related to the vortex finder length as a function of the operating Reynolds number, where the operating Reynolds number is defined in terms of the velocity of the gas-liquid phase system, inlet diameter of the proposed device and physicochemical properties of the working fluid system.

14. In addition, it would be nice if the authors could discuss the concentration field distribution of the gas-liquid mixture within the proposed device.

15. In the separation efficiency section, please explain first how to calculate gas separation efficiency using mathematical expression and liquid loss rate and then discuss the separation efficiency as a function of time and amplitude frequency, Hz.

6. PLOS authors have the option to publish the peer review history of their article (what does this mean?). If published, this will include your full peer review and any attached files.

Reviewer #1: No

Reviewer #2: No

---

## [Author Response · Author response to Decision Letter 0]

31 May 2024

We thank the reviewers for the helpful comments, which help improve the quality of the paper. Below, reviewers’ comments are presented with italics to distinguish from our response. We also submit a Word copy of the revised manuscript in which important revisions are highlighted for easy identification.

Reviewer #1: 

This article must be strictly revised by considering the following comments.

Comment 1) For abstract, the authors did not inform the reader that computational fluid dynamics was used as a tool for the authors’ work. So, the reviewer suggests the authors to add the comprehensive and informative abstract in the revised manuscript.

Response to comment 1: We thank the reviewer for the helpful comment. We have added the research tools used in the article to the abstract.

Revision: On Page 1, Paragraph 1 of the “Abstract” section, Line 4-7, the revised text reads, “Based on Computational Fluid Dynamics (CFD), Computational Solid Mechanics Method (CSM) and fluid-solid coupling method, a fluid-solid coupling mechanical model of a gas-liquid cyclone is established.”

Comment 2) From Figure 1, the reviewer understands that the hydrocyclone is horizontally located. Does the reviewer understand your considered hydrocyclone correctly? Please clearly describe your considered hydrocyclone system in the revised manuscript.

Response to comment 2: We thank the reviewer for the helpful comment. For the actual use of hydrocyclone systems, both horizontal and vertical positioning are feasible in terms of structure. In actual working conditions, the impact of centrifugal force is much greater than gravity. The description of the hydrocyclone systems is detailed in the first paragraph of section 2.1.

Comment 3) The authors only showed the schematic of the considered hydrocyclone. However, there is no information of the dimensions of the considered hydrocyclone. Please report all details of the considered hydrocyclone in the revised manuscript.

Response to comment 3: We thank the reviewer for the helpful comment. We have added information on the structural dimensions of the hydrocyclone, as shown in Figure 1b) and Table 1.

Revision: On Page 4, Paragraph 2 of section 2.1, the revised text reads, “The main size parameters of the gas-liquid separator structure are shown in Figure 1b). The main diameter D is 96mm, the number of spiral guide blades is set to 5, the length of the gas phase flow channel is L1, the inner diameter of the gas phase outlet is D1, the height of the bottom flow inverted cone is L2, the inner diameter of the liquid phase outlet is D2, the pitch is L3, the inner diameter of the bottom flow inverted cone is D3, and the length of the main body of the separator is L4. The main structural parameters of each part of the initial structure are shown in the figure 1b), and the specific dimensions of the structural parameters of each part are shown in Table 1. Each parameter is expressed as a multiple of the main diameter D.” Added new Figure 1b) and Table 1 in Section 2.1 of the article.

Comment 4) The authors used Reynolds stress turbulence model for this work. However, the pressure-strain sub-model was not reported. Please report the pressure-strain sub-model in the revised manuscript.

Response to comment 4: We thank the reviewer for the helpful comment. The pressure-strain sub-model adopts a linear strain model.

Revision: On Page 5, Paragraph 2 of the “2.2.1 Fluid governing equation” section, Line 3-4, the revised text reads, “Pressure-strain sub-model adopts linear strain.”

Comment 5) The authors must also report the selected wall functions of the present work.

Response to comment 5: We thank the reviewer for the helpful comment. The wall function uses the standard wall functionand.

Revision: On Page 7, Paragraph 1 of the “2.4 boundary condition setting” section, Line 4, the revised text reads, “the wall function uses the standard wall functionand.”

Comment 6) The authors only showed the information on an algorithm (Coupled) used in the present work. However, the information of spatial and temporal discretization schemes was absent. Therefore, in the revised manuscript, the spatial and temporal discretization schemes used in the present work must be reported. Moreover, please make the discussion between the selected set of numerical schemes of this work and the suggested numerical scheme set of the previous work (10.1016/j.powtec.2023.118713).

Response to comment 6: We thank the reviewer for the helpful comment. We have added the spatial and temporal discretization schemes used in the present work. We also referred to the literature content (10.1016/j.powtec.2023.118713). 

Revision: On Page 7, Paragraph 1 of the “2.4 boundary condition setting” section, Line 11-13, the revised text reads, “QUICK scheme with second-order precision is used for the dispersion of momentum equation, second-order upwind scheme is used for turbulent kinetic energy and dissipation rate.” Added reference [26] in the references section.

Comment 7) Please compare and show the residence time and the selected time step size of 0.005 s in the revised manuscript to confirm that the selected time step size is appropriate for the present simulations (Generally, the time step size should be shorter than 1/100 of residence time).

Response to comment 7: We thank the reviewer for the helpful comment. The characteristic size of the calculation model in this article is 30mm, with an average speed of 0.6m/s, resulting in a residence time of 0.05s. The general time step is one to two orders of magnitude smaller than the dwell time, so 0.005s is chosen as the time step in this article. In addition, under the condition of using this time step, the physical parameters of interest have reached a convergence state, as shown in the monitoring curve of the air mass flow rate at the overflow port in Fig. R1 below. Therefore, the selected time step is suitable for the current simulation. 

Figure R1 Monitoring curve of air quality flow rate at overflow port

Comment 8) The authors did not show the convergence criterion for the present simulations. Please show the information on convergence criterion in the revised manuscript.

Response to comment 8: We thank the reviewer for the helpful comment. In this article, the accuracy of the residual is controlled to 1×10-6 during simulation. During the iterative solution process, the residual of each iteration is calculated, which is the difference between the current iteration solution and the previous iteration solution. When the residual drops to a preset threshold less than a certain threshold, it is considered that the calculation has converged.

Revision: On Page 7, Paragraph 1 of the “2.4 boundary condition setting” section, Line 13-14, the revised text reads, “and the accuracy of the residual is controlled to 1x10-6.” 

Comment 9) Please represent the governing equations used in the present work in the revised manuscript.

Response to comment 9: We thank the reviewer for the helpful comment. The corresponding formulas (1)-(5) have been added, and the fluid microelements in the hydrocyclone should satisfy the conservation of mass and momentum. The mass and momentum control equations in the fluid domain can be found in formulas (1) and (2), respectively. In order to simulate the characteristics of gas-liquid separation flow field more accurately, a more accurate Reynolds stress model (RSM) is used to complete the turbulence calculation in the flow field. The pressure-strain sub-model adopts linear strain, and the expression of the Reynolds stress transport equation is shown in formula (3). The motion equation of the structure is discretized using the finite element method during fluid structure coupling analysis, taking into account the combined effect of internal and external forces in the pipe column structure. The dynamic equation of the structure is shown in formula (4).

Revision: On Page 4, the “2.2.1 Fluid governing equation “and “2.2.2 Dynamic equation of tubular string structure” section, the revised text reads, “2.2.1 Fluid governing equation

The fluid microelements in the hydrocyclone should satisfy the conservation of mass and momentum. The governing equations of mass and momentum in the fluid domain are:

 (1)

 (2)

In order to more accurately simulate the characteristics of the gas-liquid separation flow field, a more accurate Reynolds stress model (RSM) is used to complete the turbulence calculation in the flow field. Pressure-strain sub-model adopts linear strain. The expression of the Reynolds stress transport equation is:

 (3)

2.2.2 Dynamic equation of tubular string structure

The motion equation of the structure is that in the fluid-structure coupling analysis, the finite element method is used to discrete the structure, considering the joint action of the internal and external forces of the pipe string structure, and the dynamic equation of the structure is:

 (4)

” 

On Page 5, Paragraph 2 of the “2.3 Basic parameters and settings” section, Line 3-4, the revised text reads, “[24]. See Equations (5) and (6) for the coalescence kernel and fragmentation kernel functions.”

On Page 5, Paragraph 3-4 of the “2.3 Basic parameters and settings” section, the revised text reads, “

“ (5)

 (6)

”

Comment 10) The authors used mesh-based CFD model for the present work. However, the details of grid generation and figure of grid generation used in the present work were absent. Please add information of grid generation of the considered hydrocyclone in the revised manuscript.

Response to comment 10: We thank the reviewer for the helpful comment. The wall surface of the hydrocyclone is a rigid non slip wall, and its structure is linear elastic within a small deformation range. There is only static friction between the fluid and the wall of the hydrocyclone, and the remaining damping is ignored. The mesh of the structural domain is also divided into tetrahedral meshes, and the schematic diagram of the division results is shown in figure 2 a).

Revision: On Page 3, Paragraph 3 of the section 2.1, Line 1-5, the revised text reads, “The wall of the swirl separation string is a rigid non-slip wall, and the wall of the swirl separation string is linearly elastic in a small deformation range. There is only mutual static friction between the fluid and the swirl separation string, and the rest of the damping is ignored. The mesh of the fluid domain is also divided by a tetrahedral mesh. The schematic diagram of the division results is shown in the figure 2 a).”

Comment 11) In the present work, the authors did not show grid independence study, which is an important section of the CFD article, in the present manuscript. The authors must add grid independence study in the revised manuscript.

Response to comment 10: We thank the reviewer for the helpful comment. We have conducted an independence check on the grid before officially starting the simulation calculation of the article, and this section has been added to the revised draft. 

Revision: On Page 4, Paragraph 3 of the section 2.1, Line 6-14, the revised text reads, “and the independence test of the mesh is carried out. The number of mesh elements corresponding to different division levels 1 to 5 is 650736, 846224, 1035930, 1255916, 1496220, respectively. The volume fraction of gas in the bottom stream can directly reflect the water removal performance of the gas-liquid separator, so the volume fraction of gas in the bottom stream is used as the test standard. The grid independence verification is shown in figure 2b). The graphs corresponding to levels 3 to 5 show that the gas volume fraction of the underflow is relatively stable, and the final decision is based on the accuracy of the equilibrium numerical simulation and the calculation time, and the meshing level of 1035930 is adopted.”

Comment 12) For validation, the authors validated the model by considering the collection efficiency. However, the unknown parameters for the RANS equations are mean pressure and velocities. Hence, it is better to validate the present CFD model by comparing the simulated velocity profiles with the measured profiles. Please describe why the author considered collection efficiency for model validation or show the model validation by using the velocity profile comparison.

Response to comment 12: We thank the reviewer for the helpful comment. From the currently available research, it can be seen that only the effectiveness data validates the model, and there are currently no other validation data available.

Comment 13) For velocity profile comparisons, the horizontal axis labels were dimensionless radial distance (r/R). At first sight, these horizontal axis labels are seemingly proper for velocity profile comparisons. However, the velocity profiles for swirl flow inside cyclone separators are not symmetrical. Therefore, the authors must use dimensionless distance (e.g., x/R for section S2 of Figure 2) for velocity profile comparisons in the revised manuscript.

Response to comment 13: We thank the reviewer for the helpful comment. The r/R in this article is equivalent to x/R, both representing dimensionless radial distance.

Comment 14) Color map for contour of Figure 15 was not presented. Please add it into the Figure 15 of the revised manuscript.

Response to comment 14: We thank the reviewer for the helpful comment. We have added a color map for contour of figure 15((updated to figure 16 in the revised manuscript).)

Revision: Figure 16.

Comment 15) Typing errors and English grammar must be checked.

Response to comment 15: We thank the referee for the helpful comment. We have made an overall check of the manuscript to correct the Typing errors and English grammar.

Reviewer #2:

In this study, the authors used external volume force as vibrational energy to improve gas-liquid separation efficiency. However, some comments are regarding the numerical modeling of multiphase flow in gas-liquid hydrocyclone device.

Comments are given below:

Comment 1. In the Introduction section, please cite the reference related to the Computational Fluid Dynamics method (CFD). Is it related to the VOF method?

Response to comment 1: We thank the reviewer for the helpful comment. Relevant literature [18-20] has been cited in the introduction section, confirming that the method used in this article is independent of VOF

Revision: On Page 2, Paragraph 2 of the “1 Introduction” section, Line 17-26, the revised text reads, “Li S et al. [18] compared and analyzed the velocity field, pressure field and turbulent kinetic energy distribution in the variable diameter circular tube of a hydrocyclone under coupled and uncoupled conditions, and found that the influence of coupling on the flow field in the structure cannot be ignored. Xu Y et al. [19] studied the fluid-solid coupling of the spiral flow in the cyclone separator under vibration conditions, established a two-way fluid-solid coupling model of the cyclone under vibration conditions, and found that the movement of the structure caused the migration of the flow field structure. Li S et al. [20] found that the separation efficiency of the hydrocyclone under vibration coupling conditions is lower than that under uncoupled conditions.”

“

18. Li S，ZHANG J ，WANG Z C. Numerical simulation of hydrocyclone’s inner flow field under fluid structure interaction[J]. Chemical Engineering and Machinery, 2015, 42(05):706-710.

19. XU Y，ZHANG Y Y，XU D K，et al．Study on fluid-structure interaction in hydrocyclone under vibration[J]. Journal of China University of Petroleum(Edition of Natural Science)，2017，41(4):140-147．

20. Li S, Li R, Nicolleau FCGA Study on oil–water two-phase flow characteristics of the hydrocyclone under periodic excitation[J]. Chemical Engineering Research and Design, 2020, 159: 215-224.

”

Comment 2. In the introduction section, please cite the references related to the computational solid mechanics method, fluid-structure coupling theory, and fluid-structure coupling mechanical model.

Response to comment 2: We thank the reviewer for the helpful comment. We have added and cited three references related to computational so

---

## [Decision Letter · Decision Letter 1]

18 Jun 2024

PONE-D-24-11282R1Study on flow field characteristics of gas-liquid hydrocyclone separation under vibration conditionsPLOS ONE

Dear Dr. Xu,

Thank you for submitting your manuscript to PLOS ONE. After careful consideration, we feel that it has merit but does not fully meet PLOS ONE’s publication criteria as it currently stands. Therefore, we invite you to submit a revised version of the manuscript that addresses the points raised during the review process.

As previously suggested

Add a Table for model input parameters. Also mention convergence criteria.Improve quality of Figures in particular the velocity vectors (Fig. 10) so that the flow pattern becomes clear.

Further include a reference for the “PRESTO” method

We look forward to receiving your revised manuscript.

Kind regards,

Muhammad Shakaib, PhD

Academic Editor

PLOS ONE

Journal Requirements:

Reviewers' comments:

Reviewer's Responses to Questions

**Comments to the Author**

1. If the authors have adequately addressed your comments raised in a previous round of review and you feel that this manuscript is now acceptable for publication, you may indicate that here to bypass the “Comments to the Author” section, enter your conflict of interest statement in the “Confidential to Editor” section, and submit your "Accept" recommendation.

Reviewer #1: All comments have been addressed

Reviewer #2: All comments have been addressed

2. Is the manuscript technically sound, and do the data support the conclusions?

Reviewer #1: Yes

Reviewer #2: Yes

3. Has the statistical analysis been performed appropriately and rigorously? 

Reviewer #1: Yes

Reviewer #2: Yes

4. Have the authors made all data underlying the findings in their manuscript fully available?

Reviewer #1: Yes

Reviewer #2: Yes

5. Is the manuscript presented in an intelligible fashion and written in standard English?

Reviewer #1: Yes

Reviewer #2: Yes

6. Review Comments to the Author

Reviewer #1: This revised manuscript has been revised and improved properly based on the suggestions. The reviewer suggests that this manuscript can be accepted to publish by PLOS ONE. Again, please check all governing equations that are correctly written and displayed.

Reviewer #2: In the revised manuscript, the authors have successfully addressed all the comments with detailed descriptions. However, there are some questions regarding the basic parameters of the settings sections. The comments are given below:

1. In the basic parameters and settings section, please make a table related to all model input parameters and numerical methodology with convergence criteria which will be useful for readers.

2. In the context of boundary conditions, please cite the reference related to the “PRESTO” method which is used to understand the pressure distribution within the proposed system.

7. PLOS authors have the option to publish the peer review history of their article (what does this mean?). If published, this will include your full peer review and any attached files.

Reviewer #1: No

Reviewer #2: No

---

## [Author Response · Author response to Decision Letter 1]

24 Jun 2024

Response to the reviewer Comments: PONE-D-24-11282 R1

Manuscript PONE-D-24-11282 R1

 “Study on flow field characteristics of gas-liquid hydrocyclone separation under vibration conditions”

We would like to express our gratitude to the reviewers for their valuable feedback, which has contributed to further improving the quality of the paper. Below, reviewers’ comments are presented with italics to distinguish from our response. We also submit a Word copy of the revised manuscript in which important revisions are highlighted for easy identification.

This article must be strictly revised by considering the following comments.

Comment 1) i. Add a Table for model input parameters. Also mention convergence criteria.

Response to comment 1: We thank the reviewer for the helpful comment. We have added the Table 3, which provides the model input parameters and convergence criteria mentioned in this article.

Revision: On Page 6, the last paragraph of section 2.4 "Boundary condition settings".

“

The input parameters and convergence criteria involved in the calculation of this article are shown in Table 3.

Table 3. Table for model input parameters and convergence criteria

”

Comment 2) ii. Improve quality of Figures in particular the velocity vectors (Fig. 10) so that the flow pattern becomes clear.

Response to comment 2: We thank the reviewer for the helpful comment. We have rechecked the relevant figures and replaced Fig. 10 with a higher quality one.

Revision: Fig. 10

“

Fig 10. Gas phase velocity vector diagram

”

Comment 3) Further include a reference for the “PRESTO” method.

Response to comment 3: We thank the reviewer for the helpful comment. We have added the.reference[26] for the “PRESTO” method, and updated the subsequent reference numbers.

Revision: 

On Page 6, Paragraph 1 of the “2.4 boundary condition setting” section, Line 4, the revised text reads, “the pressure term[26],”.On Page 15, Paragraph 1 of the “References”, The newly added literature is “ Garrick D P, Rajagopalan R G. An Explicit Pressure-Based Algorithm for Incompressible Flows[C]. 22nd AIAA Computational Fluid Dynamics Conference. 2015: 3201.” The original reference numbers [26], [27], and [28] have been updated to [27], [28], and [29], respectively.

---

## [Editor Report · Decision Letter 2]

1 Jul 2024

Study on flow field characteristics of gas-liquid hydrocyclone separation under vibration conditions

PONE-D-24-11282R2

Dear Dr. Xu,

We’re pleased to inform you that your manuscript has been judged scientifically suitable for publication and will be formally accepted for publication once it meets all outstanding technical requirements.

Kind regards,

Muhammad Shakaib, PhD

Academic Editor

PLOS ONE
---

## [Editor Report · Acceptance letter]

3 Jul 2024

PONE-D-24-11282R2 

PLOS ONE

Dear Dr. Xu, 

I'm pleased to inform you that your manuscript has been deemed suitable for publication in PLOS ONE. Congratulations! Your manuscript is now being handed over to our production team.

Kind regards, 

on behalf of

Dr. Muhammad Shakaib 

Academic Editor

PLOS ONE